# Gold Nanocone Array with Extensive Electromagnetic Fields for Highly Reproducible Surface-Enhanced Raman Scattering Measurements

**DOI:** 10.3390/mi13081182

**Published:** 2022-07-27

**Authors:** Satoko Fujiwara, Daiki Kawasaki, Kenji Sueyoshi, Hideaki Hisamoto, Tatsuro Endo

**Affiliations:** 1Department of Applied Chemistry, Graduate School of Engineering, Osaka Metropolitan University, Osaka 599-8531, Japan; szb02114@st.osakafu-u.ac.jp (S.F.); syb02029@st.osakafu-u.ac.jp (D.K.); sueyoshi@omu.ac.jp (K.S.); hisamoto@omu.ac.jp (H.H.); 2Japan Science and Technology Agency (JST), Precursory Research for Embryonic Science and Technology (PRESTO), Tokyo 102-8666, Japan

**Keywords:** surface-enhanced Raman scattering (SERS), nanocone, plasmonics, extensive electromagnetic (EM) field

## Abstract

Surface-enhanced Raman scattering (SERS) is a technique used to distinguish the constitution of disease-related biomarkers in liquid biopsies, such as exosomes and circulating tumor cells, without any recognition elements. Previous studies using metal nanoparticle aggregates and angular nanostructures have achieved the detection of various biomarkers owing to strong hot spots and electromagnetic (EM) fields by localized surface plasmon resonance (LSPR). Although these SERS platforms enable significant enhancement of Raman signals, they still have some problems with the fabrication reproducibility of platforms in obtaining reproducible SERS signals. Therefore, highly reproducible fabrication of SERS platforms is required. Here, we propose the application of a polymer-based gold (Au) nanocone array (Au NCA), which extensively generates an enhanced EM field near the Au NCA surface by LSPR. This approach was experimentally demonstrated using a 785 nm laser, typically used for SERS measurements, and showed excellent substrate-to-substrate reproducibility (relative standard deviation (RSD) < 6%) using an extremely simple fabrication procedure and very low laser energy. These results proved that a Au NCA can be used as a highly reproducible SERS measurement to distinguish the constitution of biomarkers.

## 1. Introduction

Liquid biopsies are a powerful noninvasive tool for the early diagnosis and monitoring of various diseases. In particular, exosomes and circulating tumor cells (CTCs) are useful biomarkers for liquid biopsies. Therefore, the detection of these biomarkers is important. However, these biomarkers are composed of several heterogeneous biomolecules. For example, the composition of exosomes secreted from normal cells is different from that secreted from cancer cells [1,2], and it has been reported that CTCs have various surface expressions; thus, they have some subpopulations related to the cancer metastatic process [3,4]. Therefore, it is essential to discriminate between these different constitutions to predict cancer metastasis or invasion and to identify the primary lesion. They can commonly be detected and identified using mass spectrometry [5], fluorescence microscopy [6], and biochemical analytical tools such as enzyme-linked immunosorbent assays [7]. However, these methods have limitations. Although some of the information is obtained by the division of CTCs into their many components such as proteins and lipids, or by using several molecular recognition elements such as antibodies and aptamers, it is difficult to comprehensively determine the constituents without sample pretreatment or labelling [8,9,10]. In addition, discriminating the target biomarkers is a long and complicated process.

To overcome these problems, we focused on Raman spectroscopy. Raman spectroscopy is a vibrational spectroscopic technique that is used to obtain comprehensive information, including molecular vibrations and crystal conditions. The spectral specificity of Raman spectroscopy is recognized as a powerful tool for identifying target biomarkers. The constitution of biomarkers can be detected using Raman spectroscopy, specifically without pretreatments and recognition elements [11,12,13]. Conventional Raman spectroscopy detects only weak Raman scattering signals. To counteract this drawback, surface-enhanced Raman scattering (SERS), which enhances the Raman scattering signals, has been studied theoretically and experimentally [14].

SERS enhances Raman scattering signals using localized surface plasmon resonance (LSPR), which cannot be measured using conventional Raman spectroscopy. When the target biomarkers are close to or adsorbed by a rough metal surface, the intensity of the Raman spectra can be enhanced and detected due to electromagnetic (EM) enhancement [15,16,17]. SERS-active substrates with metal nanostructures have been studied to distinguish biomarkers such as CTCs and exosomes [18,19,20,21,22,23]. For example, Xue, T. et al. and Pang, Y. et al. developed SERS-active nanoparticles, such as gold nanoparticles, and detected CTCs due to hot spots that emerged between those nanoparticles [24,25]. Ning, C. et al. and Tian, Y. et al. also fabricated SERS-active nanoparticles similar to those for CTCs and distinguished various exosomes [26,27]. These SERS substrates exhibit extremely high sensitivity and enable the determination of the composition of biomarkers commonly found in hot spots, such as gaps between metal nanoparticle aggregates. However, certain problems remain. These hot spots are randomly distributed and difficult to reproduce accurately. Additionally, the size of the nanoparticles used in these SERS-active nanoparticles are smaller than or similar to CTCs and exosomes. Further, the added number of nanoparticles is the most important factor for detecting biomarkers precisely. The study of this adjustment is probably time-consuming. Furthermore, the high-power lasers for SERS produce extremely high photothermal temperatures, which lead to detrimental effects on biomarkers even though Raman signals are strongly enhanced [17,21,22,23,28,29,30,31]. To overcome the disadvantages of metal nanoparticle aggregates, periodic angular nanostructures have also been developed. Although the reproducibility of fabrication was improved, hot spots exist locally. Thus, many studies have reported using angular nanostructures in combination with metal nanoparticles to increase the number of hot spots. Moreover, the fabrication of these substrates, including etching and electron-beam lithography, is complicated and time-consuming [17,32,33,34,35,36]. Therefore, it is essential to develop a metal-coated substrate in which hot spots are homogeneously distributed over a wide area of the substrate by a simplified fabrication process. In addition, it is important to use a low-energy light source. Recently, various polymer-based SERS substrates such as polydimethyl siloxane (PDMS) [37,38,39], polyethylene terephthalate (PET) [40,41], polymethyl methacrylate (PMMA) [20], etc., have been developed to reduce the complexity and fabrication time. Using these polymers, previous reports achieved simplified fabrication with good reproducibility. Some of those substrates were reported to have fabricated nanopatterns using imprint lithography and self-assembly polystyrene nanoparticles. It is likely difficult to achieve good reproducibility with these SERS substrates using various mold without fabricating them in relatively large areas, because the more fabrication steps there are, the more error factors there are. However, it may be hard to fabricate the large-area SERS substrates through mold transferring and release in laboratories.

To meet these requirements, we proposed the application of a gold nanocone array (Au NCA) as a SERS substrate, which is known to generate a highly and extensively enhanced EM field near the Au NCA surface by LSPR [42,43]. In this work, the Au NCA substrate was fabricated by direct deposition of Au on a cycloolefin polymer (COP) to demonstrate the simple fabrication. Herein, we demonstrate high-throughput and reproducible fabrication of Au NCA substrates for use in SERS measurements.

## 2. Materials and Methods

### 2.1. Fabrication of Au NCA

The Au NCA was fabricated via direct deposition (Figure 1). First, a moth-eye-structured cycloolefin polymer (COP) film (FMES250/300-100×100, Scivax Co., Ltd., Kanagawa, Japan) was cut into a 30 mm square, cleaned using 2-propanol (Kanto Chemical Co. Inc., Tokyo, Japan) and ultrapure water, and then dried. Second, a Au layer (thickness: 50 nm) was thermally deposited onto the COP film. The deposition rate was adjusted in several phases to obtain better adhesion between the COP and Au and to minimize the grain boundary size as much as possible (especially in the first and final phases) following previous reports [44,45]. For the initial Au layer of 0–5 nm, the deposition rate was 0.1 nm/s. For 5–10 nm, 10–40 nm, 40–45 nm, and 45–50 nm, the deposition rates were 0.2, 0.3, 0.2, and 0.1 nm/s, respectively. Finally, the obtained Au NCA was cut into a 5 mm square (Figure 2) and fixed on glass using an adhesive sheet.

### 2.2. Microscopy

The COP film and Au NCA structure were observed using field-emission scanning electron microscopy (FE-SEM) (SU8010, Hitachi, Ibaraki, Japan) at an acceleration voltage of 10 keV. Their surfaces were also analyzed using an atomic force microscope (AFM) (AFM5000II, Hitachi, Tokyo, Japan) in tapping mode using a silicon tip cantilever (SI-DF20) with a nominal spring constant of 0.08 N/m.

### 2.3. Optical Characterization

The optical setup was composed of a tungsten halogen light source (LS-1), fiber probe (R400-7, UV-vis), spectrophotometer (USB4000), and a software program (Ocean View), all of which were purchased from Ocean Optics, Tokyo, Japan. The reflection spectra of the Au NCAs were measured in air, and then, the subtracted spectrum (1-*R*) was obtained using the normalized reflection intensity of the Au NCA (*R*) calculated by the reflection intensity of gold (thickness: 50 nm) on a glass substrate (Au glass) as reference. Optical measurements were performed in triplicate using three independent Au NCA substrates (*N* = 3). A finite difference time domain (FDTD)-method-based solution (Ansys, Inc., Vancouver, Canada) was used as the software for calculation to set the refractive index as 1 (Appendix A).

### 2.4. Evaluation of Au NCA Performance as SERS Substrate

The Au NCA substrates were immersed in various concentrations of 4-mercaptobenzoic acid (4-MBA) (>90%, Sigma Aldrich, St Louis, MO, USA) ethanolic solution (2 mL) for 3 h, washed with ethanol to remove unbound species, and dried. All Raman spectra were collected using a laser confocal Raman microscope system (RAMAN-11, Nanophoton, Osaka, Japan) with a 532 and 785 nm laser (there were not any other lasers in this Raman microscope). In this study, the SERS measurements were performed under the following conditions: 50× objective lens (N.A. = 0.8, spot diameter (*d*_spot_) = 1.2 μm), 1 mW laser power (532 or 785 nm laser), 60 s integration time, and 50 μm slit width for 4-MBA on the Au NCA and the bare Au NCA; 50× objective lens, 114 mW laser power (only 785 nm laser), 1 s integration time, and 50 μm slit width for 4-MBA on Au glass, as Au substrates without nanopatterns such as Au glass hardly generate any SERS signals under the same conditions as Au substrates with nanopatterns in general. All SERS measurements were performed using three independent substrates (*N* = 3), and SERS signals were collected at more than 15 random points from each substrate to calculate the average value.

## 3. Results and Discussion

### 3.1. Fabrication and Optical Characterization of Au NCA

The fabricated Au NCA (with a Au thickness of 50 nm) is shown in Figure 2a. A specific color (red) from the LSPR was visually observed by the naked eye and provided confirmation of successful fabrication. The surface structures of the fabricated Au NCA obtained by FE-SEM and AFM are shown in Figure 2b,c. According to Figure 2b, the diameter of the fabricated Au NCA structure was slightly smaller than that of the COP film, as shown in Appendix A. This was likely caused by the shrinking of the COP film during Au deposition. In fact, the diameter of the fabricated Au NCA was the same as that of the COP film, but they appeared different as the Au NCA was fabricated by Au deposition on this COP film (shown in Figure 1). Appendix A shows the images and cross-sectional analysis of the surface of the COP film and the Au NCA obtained by AFM. Comparing both images, the fabricated Au NCA had the same shape as the COP film. The diameter and the height of the Au NCA and the COP film shown in Appendix A were approximately 300 nm and 200 nm, respectively.

The absorption spectrum of the fabricated Au NCA (Au thickness: 50 nm) and simulated spectrum are shown in Figure 3a, where the simulated model was constructed based on the SEM and AFM images shown in Figure 2. The EM fields calculated based on the model at 532 or 785 nm are shown in Figure 3b. Comparing the experimental and simulated spectra in Figure 3a, the LSPR peak wavelengths were 532 and 572 nm, respectively. In addition, the absorbance of the fabricated Au NCA at 785 nm was larger than that of the simulation. The simulated Au NCA structure model was not exactly the same as the fabricated one; the incident light was polarized in the simulation, while it was not polarized in the experiment. This specific difference was considered the cause of the differences in the peak wavelengths and absorption intensity at both 532 and 785 nm between the experiment and simulation. These differences in the absorption spectra between the experiment and simulation have the potential to influence SERS efficiency. From the simulation results shown in Figure 3b, it was confirmed that the LSPR was excited at both 532 and 785 nm, and the EM fields at both wavelengths were spread extensively on the surface of the Au NCA. Therefore, the Au NCA could be a potential candidate for SERS substrates at both 532 and 785 nm. We first optimized the excitation laser wavelength (532 or 785 nm) and then evaluated the SERS performance.

### 3.2. Evaluation of Au NCA Performance as SERS Substrate

#### 3.2.1. Optimization of Excitation Laser Wavelength

In this study, 4-MBA was used to evaluate the Au NCA performance as a SERS substrate. We first confirmed the Au NCA SERS performance at each excitation laser wavelength (532 and 785 nm). As shown in Figure 4a, the two peaks at 1080 and 1594 cm^−1^ attributed to 4-MBA (detailed 4-MBA attribution is provided in Appendix A) were observed for both lasers after 10^−3^ M 4-MBA was adsorbed on the Au NCA. However, it showed that SERS intensity excited by a 785 nm laser was approximately 60 times larger than when excited by a 532 nm laser. In general, the SERS enhancement mechanism can be explained by the strength of the EM field [15,16,17,46,47]. In the case of this Au NCA (Au thickness: 50 nm), the maximum strengths of EM fields (|*E*_max_/*E*_0_|) at the 532 and 785 nm wavelengths were about 4.99 and 4.37, respectively. Thus, the SERS intensity that we would theoretically expect would be more excited by 532 nm than by 785 nm, if both EM field distribution were formed similarly. However, the experimental results are different. This might have been caused by the EM field distribution at each excitation laser. As shown in Figure 3b, the EM field at 785 nm was mostly distributed on the surface of the Au NCA, whereas the EM field at the 532 nm laser was distributed not only on the surface but also inside the Au NCA. Thus, the absorption intensity (1-*R*) of this Au NCA at 532 nm was larger than that at 785 nm, as the EM field expanded on the inside polymer of the Au NCA. These results suggested that the plasmon supplied to 4-MBA on the surface of the Au NCA when it was excited by the 532 nm laser might be less than that when it was excited by the 785 nm laser (Figure 4b). When this core–shell (polymer-Au) Au NCA was excited by the 532 nm laser (which is a smaller frequency than the plasma frequency of gold material [48]), part of the incident light may have been absorbed in the Au itself, the surface of the Au NCA, and the gap between the Au and the polymer and transmitted on the inside of the Au NCA (polymer). This Au NCA feature has already been reported in our previous reports [42,43]. That is likely why the SERS intensity shown in Figure 4a was different. Additionally, we would like to emphasize that the 532 nm laser is never unsuitable for SERS detection on SERS substrates made of gold material. Some previous studies have reported fabricating the SERS substrates made of gold material, and demonstrated SERS detection by 532 nm [49,50]. This Au NCA was only unsuitable for the detection of the SERS signal excited at 532 nm for the above reasons. That is why we chose the 785 nm laser in this study and used it after the next evaluation.

#### 3.2.2. Theoretical Consideration of the SERS Enhancement Effect of Au Thickness on Au NCA by FDTD Simulation

As mentioned above, the SERS enhancement is mainly explained by the strength of the EM field. Therefore, each of the EM fields at 785 nm with various thicknesses of the Au layer (40, 50, 60, 80, and 100 nm) on NCA were also investigated by FDTD simulations. As shown in Appendix A, the EM field largely covers the Au NCA surface for each Au thickness. Appendix A shows the maximum values of the EM field intensities (|*E*_max_/*E*_0_|) on the Au NCA with various thicknesses of the Au layer. |*E*_max_/*E*_0_| decreased slightly as the thickness of the Au layer increased and the absorption intensity (1-*R*) decreased, as shown in Appendix A.

The Au NCA structures used in this study, namely moth-eye structures, are close to flat structures with increasing Au thickness. That is to say, the surface area of the Au NCA decreases and the aspect ratio of this Au NCA becomes smaller when increasing the Au thickness as the gaps between the nanocones (the blue line in Appendix A) are also covered. This is expected to cause a loss in SERS potential. Therefore, this Au NCA work obtained opposite experimental results compared to other reports [51,52]. |*E*_max_/*E*_0_| was largest at a thickness of 40 nm. However, the EM field expanded inside of the Au NCA when the Au layer was 40 nm, which was expected to lead to a loss in plasmonic energy, as mentioned above. Thus, we chose a thickness of 50 nm and then experimentally evaluated the sensitivity and reproducibility of this Au NCA at 785 nm.

#### 3.2.3. Evaluation of the Sensitivity and Reproducibility of Au NCA SERS Substrate

As shown in Figure 5a, SERS signals of 4-MBA on this Au NCA (Au thickness: 50 nm) could be clearly identified even at 10^−6^ M. At 10^−7^ M or less; 4-MBA could not be detected, and the Au NCA signals were observed instead. This might be caused by the extensive EM field distribution on the Au NCA surface. The bare Au NCA area in the laser spot increased as the number of 4-MBA molecules decreased. This was also observed in the case of 10^−6^ M 4-MBA on Au NCA (Figure 5b).

The SERS enhancement factor (EF) of the Au NCA was investigated to quantitatively evaluate the Raman signal enhancement according to Equation (1):EF = (*I*_SERS_/*I*_NR_)/(*N*_SERS_/*N*_NR_)(1)
where *I*_SERS_ and *I*_NR_ correspond to the Raman intensity of 4-MBA on the Au NCA and 50 nm Au glass, respectively (Appendix A). *N*_SERS_ and *N*_NR_ refer to the number of 4-MBA molecules on the Au NCA (10^−6^ M) and Au glass (10^−2^ M), respectively. Thus, the EF was estimated to be approximately 1 × 10^5^ based on the characteristic peaks at 1080 and 1594 cm^−1^ (details of the calculation are described in the Appendix A). Finally, we evaluated the spot-to-spot and substrate-to-substrate reproducibility for the Au NCA. The spot-to-spot reproducibility of the same Au NCA substrate is shown in Figure 6. All of the spots showed a relatively consistent Raman intensity. The relative standard deviations (RSDs) of the Raman intensities at 1080 and 1594 cm^−1^ were calculated as 7.98% and 8.67%, respectively (Figure 6b). Concerning the substrate-to-substrate reproducibility of the SERS signal, the RSDs were 5.46% and 5.81% at 1080 and 1594 cm^−1^, respectively (Figure 6c).

We compared these results for this Au NCA to those of previous reports that examined different nanocone SERS substrates fabricated by a polymer (Table 1). At first, the limit of detection (LOD) and the EF value for the Au NCA were comparable to or less than those of previous reports. This was caused by the Au NCA structures. As shown in Figure 3b, this Au NCA had extensive EM field distributions on the surface of the Au NCA, but there were fewer areas of extreme concentration of EM fields (such as hotspots) on this Au NCA compared to previous reports. Thus, the LOD was higher and the EF was smaller for this Au NCA. However, even without hotspots, this Au NCA had enough potential to generate SERS, as mentioned above. This indicated that many hot spots are not always necessary to generate SERS if there are medium strength and extensive EM fields. Further, this Au NCA achieved excellent substrate-to-substrate reproducibility (RSD). Because these small substrate-to-substrate RSD values were rarely obtained using previously reported methods, our approach of using polymer film cut as large as possible (where the limit was about 30 mm square in this study), followed by Au deposition is quite advantageous for obtaining excellent reproducibility. Those results were similar to those reported by Vignesh, S. et al., [51]. However, this Au NCA needed much lower laser power compared to that of Vignesh, S. et al. [51]. Lower laser energy will prevent photothermal effects that are detrimental to biomarkers.

Finally, we compared our fabrication process with that of the other nanocone SERS substrates shown in Table 1. These nanocone SERS substrates are needed to fabricate some molds for polymer transfer by either nanoimprint lithography (heat [51] or UV [54]), etching (ion [52], or anodization and chemical [53]), or both. Compared to those methods, our fabrication method was simpler; only gold deposition on COP film was needed, as we did not need to fabricate any molds for polymer transfer.

This Au NCA was advantageous in terms of fabrication simplicity, extreme reproducibility, and very low laser energy needed to obtain SERS measurements. In addition, it was also found that this Au NCA overcame the problems in former studies using metal nanoparticle aggregates and angular plasmonic nanostructures.

## 4. Conclusions

The Au NCA was successfully fabricated using a very simple process. From the simulation analysis, the EM field was expanded at the Au surface. Considering the SERS performance, this Au NCA achieved good sensitivity (LOD ≈ 1 μM, EF ≈ 10^5^) and excellent reproducibility (RSD < 6%). Although the EF value of the Au NCA was comparable with or smaller than that of previous studies, as there were few areas of extreme concentration of EM fields such as hotspots at this Au NCA, this Au NCA had enough potential to generate SERS with low laser energy because it had medium strength and extensive EM fields. In addition, the substrate-to-substrate RSD the was small for this Au NCA due to the simple fabrication thorough direct Au deposition at the polymer. These results prove that the Au NCA substrate can be used as a SERS substrate and can be applied in several fields, such as life sciences and pharmaceutical sciences. The fabrication of SERS substrates with higher EF values should be further investigated in future work by changing the types of metal coatings, etc. We expect this Au NCA to be useful for distinguishing biomarkers, such as exosomes and CTCs.

## Figures and Tables

**Figure 1 micromachines-13-01182-f001:**
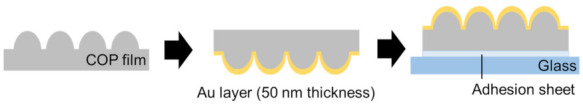
Fabrication procedure of Au NCA using direct deposition.

**Figure 2 micromachines-13-01182-f002:**
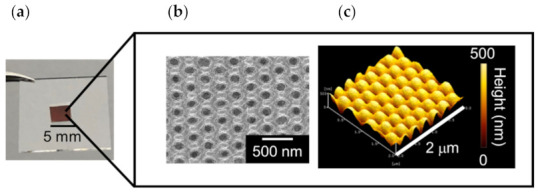
Images of the Au NCA (**a**) and its surface structure as observed by FE-SEM (**b**) and AFM (**c**).

**Figure 3 micromachines-13-01182-f003:**
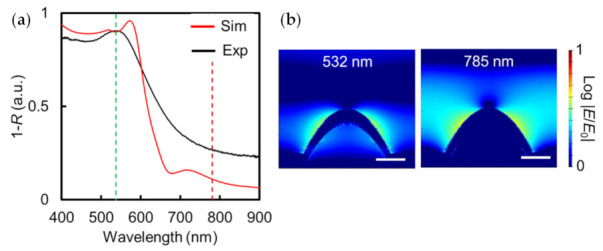
(**a**) Absorption spectra of Au NCA (Au thickness of 50 nm) in the experiment (black line) and simulation (red line). The green and dark red dashed lines show the position at 532 nm and 785 nm wavelength, respectively. (**b**) EM field distributions of a Au NCA cross-section at 532 nm or 785 nm in the simulation (white scale bar is 100 nm).

**Figure 4 micromachines-13-01182-f004:**
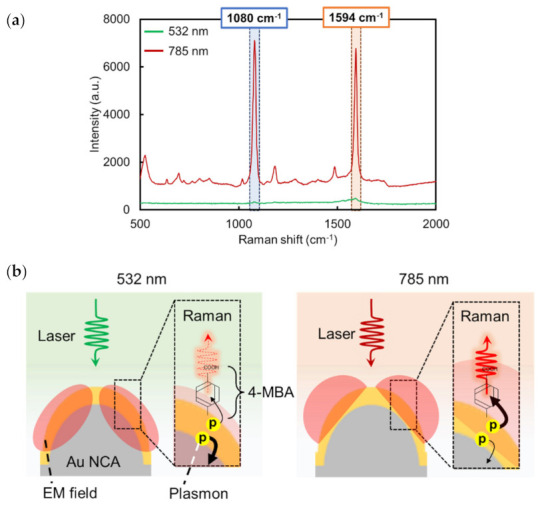
(**a**) Raman spectra of 10^−3^ M 4-MBA on Au NCA (Au thickness of 50 nm) excited by a 532 or 785 nm laser. (**b**) Illustrations of EM field distributions of Au NCA excited by 532 or 785 nm.

**Figure 5 micromachines-13-01182-f005:**
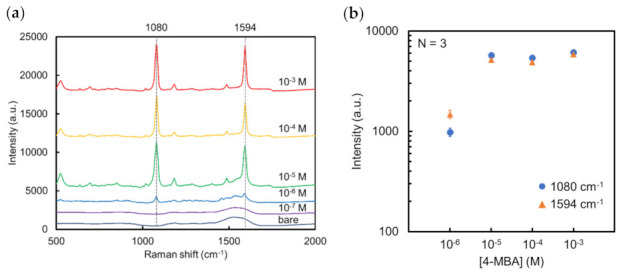
(**a**) SERS spectra of 10^−7^–10^−3^ M 4-MBA on Au NCA and bare Au NCA. (**b**) Concentration dependency for 4-MBA at 1080 and 1594 cm^−1^.

**Figure 6 micromachines-13-01182-f006:**
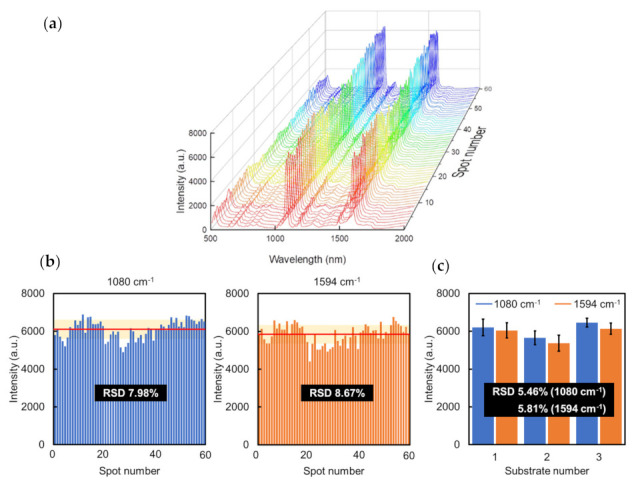
(**a**) SERS spectra of 4-MBA (10^−3^ M) randomly collected at 60 sites from three Au NCA substrates; (**b**) spot-to-spot intensity distribution at 1080 and 1594 cm^−1^ (left and right, respectively. the average intensity is indicated by the red line, and the light-yellow zones represent the ±RSD; (**c**) substrate-to-substrate intensity distribution at 1080 and 1594 cm^−1^.

**Table 1 micromachines-13-01182-t001:** Comparison of the limit of detection (LOD), enhancement factor (EF), and reproducibility (RSD) for different nanocone SERS substrates fabricated by polymers.

SERS Substrate	SERS Condition	LOD	EF	RSD
Au NCA (This work)	785 nm(1 mW, 60 s)	1 μM	1.15 × 10^5^	5.46%
Nanocone polycarbonate [51]	785 nm (35 mW, 5 s)	1 μM	1 × 10^5^	- *
Au-capped polymer nanocones [52]	780 nm (5 mW, 0.05 s)	0.1 μM	5 × 10^6^	- *
Polymer-nanocone-based 3D Au nanoparticle array [53]	532 nm (5 mW, 10 s)	1 pM	1.3 × 10^8^	<11%
Au covered polymer nanostructure array [54]	785 nm (1 mW, 10 s)	- *	1.21 × 10^7^	7.93%

* The authors did not mention.

## Data Availability

The data are contained within the article and Appendix A.

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
