# Peer review of "Gold Nanocone Array with Extensive Electromagnetic Fields for Highly Reproducible Surface-Enhanced Raman Scattering Measurements"

_micromachines, 2022, doi:10.3390/mi13081182_

Round 1

Reviewer 1 Report

In the manuscript entitled "Gold Nanocone Array with Extensive Electromagnetic Fields for Highly-Reproducible Surface-Enhanced Raman Scattering Measurements", S. Fujiwara et al. describe the fabrication of arrays on gold nanocones by a simple method consisting in the metallization of a pre-structured template, and analyse Raman (SERS) measurements obtained using such a substrate.

The principle of the method is sound since it excludes lithography processes, and provides arrays of nanocones with ~300 nm pitch.

The paper however lacks from a series of imprecisions/issues that does not allow it to be published, at least in the present form.

- the individual cone apex radii have not been provided in the manuscript, while they should drive a priori the efficiency of the structure for SERS measurements. Such data can be extracted from AFM measurements (such as data shown in the supp info), and even a statistics of this parameter should be obtained as a testbench of the fabrication process reliability.

- how does the fabrication process compare to literature ? Only a few references are provided, i.e. Refs [17, 28-32], including a very general review of SERS substrates. This does not necessarily specifically compare with a review of metal nanocone fabrication processes. This point should be improved to estimate the originality or performance of the fabrication scheme.

- I am surprised of the relatively modest agreement between the simulated and experimental absorption spectra in Figure 2, given the fact that the fabricated structures are typicall sub-micron in size (rather than actually nanometric). Hence, a full geometric description of the sample should be sufficient to match with experimental data. One possibility offered by the authors to account for the disagreement is the use of non-polarized versus polarized lighting : this could/should be easily discriminated, either from the simulations or from experiments. Also, it is said that the simulated structure is not exactly the experimental one : why is this so, given the simple fabrication method and structural data available in order to achieve an accurate description of the experimental sample ?

- the inversion of the expected SERS enhancements at 532nm and 785nm is explained as from the simulations of electric fields at the top/close to the top of the Au tips. Since the authors explain that the simulations are based on a different geometry s compared to the real samples, how robust is this argument as for the explanation of the inversed SERS enhancements at 532nm and 785nm ?

- the Au cones should have an apex radius larger than 50nm (diameter of about 100nm, as deduced from the simulations in Fig. 3b). This is actually fairly large (and again a comparison with litterature would be welcome). In particular, any disorder / granularity of the metal coating (at as smaller scale) are likely to provide with more hotspots. Can the authors measure such a granularity e.g. from AFM data, and link it (or not) with the reproducibility of SERS measurements ?

In conclusion, while the authors provide an interesting lithographic-free method for the fabrication of SERS substrates,  the manuscript should definitely be improved in view of publication.

Author Response

Response to the Reviewer 1

Thank you very much for your comments and advice. Our comments to your revisions are following.

  1. - the individual cone apex radii have not been provided in the manuscript, while they should drive a priori the efficiency of the structure for SERS measurements. Such data can be extracted from AFM measurements (such as data shown in the supp info), and even a statistics of this parameter should be obtained as a testbench of the fabrication process reliability.

Thank you for your advice. Your advice is definitely right.

At first, we apologized that the basic structure information of the fabricated Au NCA was lacked in the main text. The diameter and height of the COP film and the fabricated Au NCA shown in Figure S3 were about 300 nm and 200 nm, respectively. We added this structure information in the main text and please see the highlighted line 158−160 in the main text (page 4).

Then, the individual cone apex radii as you pointed out were calculated by the following steps.

 1. The cross-sectional shape numeric data on a blue line (three cones) in AFM images (Figure S3a

and b) was approximated by a quadratic curve (y = ax2 + bx + c; a, b and c are coefficient).

(The nanocones on the blue line were chosen to be consistent with the structure of the simulation.

The fabricated Au NCA had the caves, whereas the simulated one had not any caves, as we also

mentioned the following answer to your question 3.)

  1. The curvature at the individual cone apex of each cone is approximately determined by 1/2|a|.

Following the above steps, we calculated the individual cone apex radii. As a result, it was about 74.3 nm (the COP film) and 77.4 nm (the fabricated Au NCA), respectively, as you also assumed below the suggestion 5. The individual cone apex radii of the fabricated Au NCA were larger than the COP film. This is probably caused to use the direct Au deposition on COP film in fabrication process.

Thanks to your advice, we added to write this individual cone apex radii of both in the supplementary information (section: Evaluation of the individual cone apex radii of the fabricated Au NCA and the COP film) and please see the relevant part.

  1. - how does the fabrication process compare to literature ? Only a few references are provided, i.e. Refs [17, 28-32], including a very general review of SERS substrates. This does not necessarily specifically compare with a review of metal nanocone fabrication processes. This point should be improved to estimate the originality or performance of the fabrication scheme.

Thank you for your advice.

Your advice is appropriate. We added the description compared with other nanocone SERS substrates about the fabrication process and please see the other references and the highlighted line 294−299 in the main text (page 8−9).

  1. - I am surprised of the relatively modest agreement between the simulated and experimental absorption spectra in Figure 2, given the fact that the fabricated structures are typicall sub-micron in size (rather than actually nanometric). Hence, a full geometric description of the sample should be sufficient to match with experimental data. One possibility offered by the authors to account for the disagreement is the use of non-polarized versus polarized lighting : this could/should be easily discriminated, either from the simulations or from experiments. Also, it is said that the simulated structure is not exactly the experimental one : why is this so, given the simple fabrication method and structural data available in order to achieve an accurate description of the experimental sample ?

Thank you for your comments and question. Your comments and question are appropriate.

As you pointed out, the polarized incident light was irradiated, which was different from the experiment, in this simulation. However, the optical properties were expected to be characterized because Au NCA was isotropic and LSPR mode did not depend on the lattice structure as our previous study reported [42,43]. Therefore, the shape of the simulated and experimental absorption spectra was roughly similar.

Although the structures are matched with the fabricated Au NCA and the simulated one (diameter: 300 nm, height: 200 nm), the simulated and experimental absorption spectra were not completely identical, in particular, the peak wavelength. Then, we thought that this difference was caused by the caves between nanocone and nanocone where is the black oval places in Figure S3. Therefore, we have previously tried to form these caves in the simulation structures. However, that simulation result added the caves structures was more apart from the experimental one about the shape and the peak wavelength. Following this result, it was decided to mimic this cave by reducing the distance from the Au plate surface to the Au NCA tip. Thus, the peak wavelength of the simulation was the most similar to the experimental one when the bottom of the nanocone structure was 100 nm below the surface of the Au plate. That’s why we said that the simulated structure is not exactly the experimental one.

Thanks to your polite suggestion, we added that explanation and references about the simulation in the supplementary information and please see that highlight sentences (section: Explanation about FDTD simulation in this study).

  1. - the inversion of the expected SERS enhancements at 532nm and 785nm is explained as from the simulations of electric fields at the top/close to the top of the Au tips. Since the authors explain that the simulations are based on a different geometry s compared to the real samples, how robust is this argument as for the explanation of the inversed SERS enhancements at 532nm and 785nm ?

Thank you for your question.

Firstly, 532 nm is longer wavelength (smaller frequency) than the plasma frequency (approximately 520 nm). Thus, gold material itself has absorption in plasma dynamics when it was excited by 532 nm laser. However, plasmons may be strongly excited by the incident light and part of the incident light may be also transmitted into this Au NCA at the same time. Therefore, the EM fields were emerged not only at the surface of the Au NCA but also at the gap between polymer core and Au shell in this Au NCA with 50 nm Au thickness excited by a 532 nm laser. This was supported by our previous report [42], which is used the almost similar nanocone structure.

Then, the optical property of the gold material itself at 785 nm is mainly reflection. Also, this Au NCA structure resembles a gold flat plate with a significantly increased surface area because it is close to flat structures with increasing the Au thickness. This was also supported by our previous report [42]. According to these points, it is considered that the EM field was only spread at the surface of Au NCA excited by a 785 nm laser.

In addition, Kalachyova, Y. et al., have reported that the SERS signals and intensities stems from the different plasmon intensity distribution at 532 nm and 785 nm.

In summary, we thought the inversed SERS enhancements at 532 nm and 785 nm were strongly supported by the LSPR characterization of this Au NCA structure, plasma dynamics, our previous reports and Kalachyova, Y. et al., study.

Kalachyova, Y.; Mares, D.; Jerabek, V.; Zaruba, K.; Ulbrich, P.; Lapcak, L.; Svorcik, V.; Lyutakov, O.; The Effect of Silver Grating and Nanoparticles Grafting for LSP−SPP Coupling and SERS Response Intensification, J. Phys. Chem. C 2016, 120, 10569−10577. https://doi.org/10.1021/acs.jpcc.6b01587

Thanks to your question, we added the above explanation in the main text please see these references and the highlighted line 206−217 (page 5).

  1. - the Au cones should have an apex radius larger than 50nm (diameter of about 100nm, as deduced from the simulations in Fig. 3b). This is actually fairly large (and again a comparison with litterature would be welcome). In particular, any disorder / granularity of the metal coating (at as smaller scale) are likely to provide with more hotspots. Can the authors measure such a granularity e.g. from AFM data, and link it (or not) with the reproducibility of SERS measurements ?

Thank you for your suggestion.

Your suggestion is significantly right but it is very difficult to measure the disorder or granularity of the metal coating accurately from AFM data. However, we were careful to the disorder or granularity of metal coating as much as possible because these are related to the SERS enhancement and the reproducibility of SERS measurements as you also pointed out.

We have already checked the relationship between the deposition rate and the metal grain boundary size from some references ([44,45]) and then determined the described deposition rate. Please see Figure 2a in reference 44, the grain boundary size of Ag is slightly increasing with the deposition rate of Ag is increasing. Additionally, the Au NCA structures used in this study, namely moth-eye structures, are close to flat structures with increasing the Au thickness. The Au NCA structures might loss its original structure if the first and last phases of the deposition rate are rapid and large Au grains covered on nanocones made of COP completely. Therefore, we performed the described deposition rate.

Although we tried to reduce the roughness or disorder of metal coating as less as possible, some disorder or granularity of the metal coating through Au deposition process might be provided with lots of hotspots and be prompted SERS enhancement as you pointed out.

Reviewer 2 Report

The manuscript presents a polymer-based gold nanocone array (AuNCA) that can extensively generate an enhanced EM field on the Au surface by LSPR. They selected the Au thickness on AuNCA and excitation laser wavelength to realize highly-reproducible SERS measurements. It sounds like interesting. However, several important points should be stated clear and improved.

There are some comments as follow:

1.              The introduction part did not mention the recent progress about the polymer-based SERS active substrates, such as PDMS, PET polymer.

2.              The research target is CTCs and the detection technology is SERS. The authors were encouraged to describe the previous reports about the SERS detection of CTCs.

3.              Is near-EM field the official abbreviation of EM field neared surface? If not, please explain its theoretical meaning.

4.              The innovation of this work is to fabricate the homogeneous substrate with an extremely simple fabrication procedure and very low laser energy, but less mention advantages of its fabrication procedure. The authors are encouraged to provide the experimental result at different deposition conditions to prove the description on page 2 “The deposition rate was adjusted for several phases to get better the adhesion between COP and Au and to minimize the grain boundary size as much as possible (especially first and final phases).”

5.              This work employed EM field intensity distribution of different Au thickness. Regarding other reports, especially reference 37, 38, it may reveal sharp gaps by increasing the thickness of deposited Au in a certain range, thus enhance EM field. It seems that this work has got the opposite experiment resultsFigure.S4 (c). Please, confirm and state clearly in the text.

6.              Figure 2b)(c and  Figure S2 seem to be the same picture, please replace one of them, and the page 3 According to Figure 2b, the diameter of the fabricated Au NCA structure is slightly smaller than that of the COP film, as shown in Figure S2 is not clear.

7.              Page 4, The simulated Au NCA structure model was not exactly the same as the fabricated one, and the incident light was polarized in the simulation, while it was not polarized in the experiment. This difference was considered to cause the difference in the peak wavelength and absorption intensity at both 532 and 785 nm between the experiment and simulation. Please find references support and verify this conclusion, or describe Figure S1 in detail.

8.              Page 5, “whereas the EM field at 532 nm laser was distributed not only at the surface but also the inside of Au NCA. Thus, the absorption 1intensity (1-R) of this Au NCA at 532 nm was larger than at 785 nm because EM field was expanded at the inside polymer of Au NCA. These results suggested that plasmon supplied to 4-MBA on the surface of Au NCA when it was excited by 532 nm laser might be less than when it was excited by 785 nm.”. Please give a strong theoretical support from the perspective of plasma dynamics. The 532 nm laser is usually not suitable for gold materials in SERS detections, why the authors selected it as comparison.

9.              Additional information about the Raman fingerprinting of the SERS reporter 4-MBA, band assignment may be included in the SI.

Author Response

Response to the Reviewer 2

Thank you very much for your comments and advice. Our comments to your revisions are following.

  1. The introduction part did not mention the recent progress about the polymer-based SERS active substrates, such as PDMS, PET polymer.

Thank you for your advice.

We added this description of previous reports about polymer-based SERS substrates in the introduction and please see the highlighted line 82−91 in the main text (page 2).

  1. The research target is CTCs and the detection technology is SERS. The authors were encouraged to describe the previous reports about the SERS detection of CTCs.

Thank you for your advice.

We thought that we showed some previous reports about the SERS detection of CTCs in references 18, 21, 22 and 23. However, we added the specific examples (references 24−27) thanks to your advice. We made and added this change the wording, and please see these references and the highlighted line 59−73 in the main text (page 2).

  1. Is “near-EM field” the official abbreviation of “EM field neared surface”? If not, please explain its theoretical meaning.

Thank you for your polite question. Your suggestion is right.

As you pointed out, “near-EM field” means “EM field neared surface of Au NCA”. We made some changes the wording and please see the highlighted lines 20−21 (page 1) and 93−94 (page 2) in the main text.

  1. The innovation of this work is to fabricate the homogeneous substrate with an extremely simple fabrication procedure and very low laser energy, but less mention advantages of its fabrication procedure. The authors are encouraged to provide the experimental result at different deposition conditions to prove the description on page 2 “The deposition rate was adjusted for several phases to get better the adhesion between COP and Au and to minimize the grain boundary size as much as possible (especially first and final phases).”

Thank you for your advice. Your advice is very appropriate.

We have already checked the relationship between the deposition rate and the metal grain boundary size from some references ([44,45]) and then determined the described deposition rate. Please see Figure 2a in reference 44, the grain boundary size of Ag is slightly increasing with the deposition rate of Ag is increasing. Additionally, the Au NCA structures used in this study, namely moth-eye structures, are close to flat structures with increasing the Au thickness. The Au NCA structures might loss its original structure if the first and last phases of the deposition rate are rapid and large Au grains covered on nanocones made of COP completely. Therefore, we performed the described deposition rate. Thanks to your advice, we added these references ([44][45]) about the relationship between the deposition rate and the metal grain boundary size and please see line 106−107 in the main text (page 3).

  1. This work employed EM field intensity distribution of different Au thickness. Regarding other reports, especially reference 37, 38, it may reveal sharp gaps by increasing the thickness of deposited Au in a certain range, thus enhance EM field. It seems that this work has got the opposite experiment results(Figure.S4 (c)). Please, confirm and state clearly in the text.

Thank you for your suggestion.

The Au NCA structures used in this study are close to flat structures with increasing the Au thickness as mentioned above 4. That is to say, the surface area of Au NCA is decreasing and the aspect ratio of this Au NCA is smaller by increasing the Au thickness because gap place between nanocone and nanocone (where is on a blue line in Figure S3) is also covered. This is expected to cause the loss of the SERS potential. Therefore, this Au NCA work has got the opposite experiment results compared to other reports as you pointed out (please be careful we changed these references number about other reports, [37] to [51] and [38] to [52]). We added above explanation in the main text and please see the highlighted line 232−237 (page 6).

  1. Figure 2(b)(c) and Figure S2 seem to be the same picture, please replace one of them, and the page 3 “According to Figure 2b, the diameter of the fabricated Au NCA structure is slightly smaller than that of the COP film, as shown in Figure S2” is not clear.

Thank you for your questions.

First, Figure 2b and c, and Figure S2 are not the same picture. Figure 2b and c (the Au NCA observed by FE-SEM and AFM), and Figure S2 (the COP film observed by FE-SEM and AFM) are mostly the same structures because the COP film has NCA structures and Au NCA is fabricated by Au deposition on this COP film (shown in Figure 1).

Then, your suggestion is right. Certainly, both AFM images of Au NCA and COP film (Figure S3a and S3b) are very similar. However, it looked like the fabricated Au NCA structure is slightly different to the COP film compared to both SEM images (Figure 2b and Figure S2a). Therefore, we thought the reason why is the shrinking of the COP film during Au deposition (line 153−154 in the main text, page 4). In fact, the diameter of the fabricated Au NCA is same to that of the COP film but they are looked like different because Au NCA is fabricated by Au deposition on this COP film (shown in Figure 1). We wanted to explain and emphasized these things that it looked like the difference structures both SEM images of Au NCA and COP film but they are similar as shown in both AFM images. Thanks to your suggestion, we added the explanation in the main text and please see the highlighted line 154−156 (page 4).

  1. Page 4, “The simulated Au NCA structure model was not exactly the same as the fabricated one, and the incident light was polarized in the simulation, while it was not polarized in the experiment. This difference was considered to cause the difference in the peak wavelength and absorption intensity at both 532 and 785 nm between the experiment and simulation.” Please find references support and verify this conclusion, or describe Figure S1 in detail.

Thank you for your careful suggestion. Your suggestion is appropriate.

We added that explanation and references about the simulation in the supplementary information and please see that highlight sentences (section: Explanation about FDTD simulation in this study).

  1. Page 5, “whereas the EM field at 532 nm laser was distributed not only at the surface but also the inside of Au NCA. Thus, the absorption 1intensity (1-R) of this Au NCA at 532 nm was larger than at 785 nm because EM field was expanded at the inside polymer of Au NCA. These results suggested that plasmon supplied to 4-MBA on the surface of Au NCA when it was excited by 532 nm laser might be less than when it was excited by 785 nm.”. Please give a strong theoretical support from the perspective of plasma dynamics. The 532 nm laser is usually not suitable for gold materials in SERS detections, why the authors selected it as comparison.

Thank you for your variable comments.

Firstly, reviewer would like to think the 532 nm laser is not suitable for gold materials in SERS detections because 532 nm is longer wavelength (smaller frequency) than the plasma frequency (approximately 520 nm). Thus, gold material itself has absorption in plasma dynamics when it was excited by 532 nm laser as you pointed out. However, it is not the only possible phenomenon in LSPR. At the same time, plasmons may be excited by the incident light and part of the incident light may be transmitted in nanostructures. This phenomenon is also occurred on this Au NCA. According to our previous report [42], the EM field was emerged not only at the surface of the Au NCA but also at the gap between polymer core and Au shell because part of the incident light may be transmitted, but this Au NCA as DNA sensor high sensitivity was achieved well enough. Also, our following study [43] has reported the improvement of sensor performance using this absorption and transmittance loss occurred on this Au NCA. Therefore, we described these features using Figure 4b.

Secondly, there are some previous reports in SERS detections using gold material and 532 nm laser. For examples, Zhao, W. et al., [53] shown in Table 1 developed polymer-nanocone-based 3D Au nanoparticle array. Their developed SERS substrates had the absorption peak at approximately 560 nm but was excited by 532 nm laser. Also, other similar reports have reported in following two references please see those ([49, 50]). According to those reports, it could be said that SERS substrates made of gold material are generally unsuitable for 532 nm excitation.

[49] Ameer, S. F.; Pittman, U. C. Zhang, D. Quantification of Resonance Raman Enhancement Factors for Rhodamine 6G (R6G) in Water and on Gold and Silver Nanoparticles: Implications for Single-Molecule R6G SERS. J. Phys. Chem. C 2013, 117, 27096−27104. https://doi.org/10.1021/jp4105932

[50] Shorie, M.; Kumar, V.; Kaur, H.; Singh, K.; Tomer, K. V.; Sabherwal, P. Plasmonic DNA hotspots made from tungsten disulfide nanosheets and gold nanoparticles for ultrasensitive aptamer-based SERS detection of myoglobin. Microchimica Acta 2018, 185, 158. https://doi.org/10.1007/s00604-018-2705-x

Finally, we expected this Au NCA was more SERS potential excited by 532 nm than 785 nm from the above second reason, the SERS enhancement mechanism as we mentioned in the main text line 192−195, and the Raman intensity is generally proportional to the fourth power of the frequency of the incident light. However, the experimental results are differed from our expectations. Therefore, we demonstrated to compare excited by 532 nm laser with excited by 785 nm laser and considered these results were caused because of the above first reason.

That’s why we confirmed SERS performance using both 532 and 785 nm lasers.

Thanks to your careful remarks, we added more detail explanation and please see the highlighted line 206−217 in the main text (page 5).

  1. Additional information about the Raman fingerprinting of the SERS reporter 4-MBA, band assignment may be included in the SI.

Thank you for your advice. Your advice is appropriate.

We added the information about the Raman band assignment of 4-MBA in supplementary information (Table S1) and the related explanation line 192 in the main text (page 5).

Round 2

Reviewer 2 Report

The authors has addressed most of my comments.